# Attitudes and Beliefs of a Sample of Australian Dog and Cat Owners towards Pet Confinement

**DOI:** 10.3390/ani13061067

**Published:** 2023-03-15

**Authors:** Jacquie Rand, Zohre Ahmadabadi, Jade Norris, Michael Franklin

**Affiliations:** 1School of Veterinary Science, The University of Queensland, Gatton, QLD 4343, Australia; 2Australian Pet Welfare Foundation, Kenmore, QLD 4069, Australia

**Keywords:** pet cat, pet dog, stray, confinement, containment, animal shelter, semi-owned cat, sterilization, mental health, wildlife

## Abstract

**Simple Summary:**

In Australia, most cats and dogs entering animal shelters and pounds are classed as strays, typically from low-income areas. Most dogs and an unknown proportion of stray cats are escaped or wandering owned pets. Many of these stray animals are euthanized, negatively impacting the mental health of staff involved, and increasing the risk of depression, traumatic stress, substance abuse and suicide. Modern sheltering practices aim to reduce the number of stray cats and dogs, which reduces shelter admissions and euthanasia. Australian pet owners (*n* = 2103) were surveyed about their attitudes towards four types of pet confinement. Dog owners showed the strongest support for confining dogs to the owners’ property whenever unsupervised and less support for confining dogs inside the house at night (54% agreed), while 23% believed dogs had a negative impact on wildlife. Cat owners showed the strongest support for confining cats inside at night. Cat owners’ non-supportive attitudes towards cat confinement were partly because of higher concern for cat quality of life and lower concern about their cats’ predation behaviors, compared to non-cat owners. These results provide valuable information to inform more effective strategies to reduce stray animals and associated issues.

**Abstract:**

Most cats and dogs entering Australian animal shelters and municipal facilities are classed as strays, typically from low socio-economic areas. Contemporary practices increasingly focus on proactively reducing the number of stray animals, which requires further understanding of factors associated with straying animals, including pet confinement. Australian cat and dog owners (*n* = 2103) were surveyed to investigate attitudes towards four types of pet confinement and how these were influenced by social norms, demographics and concerns about pet quality of life and potential wildlife predation. Dog owners showed the strongest support for confining dogs to the owners’ property whenever unsupervised (98% agreement) and less support for confining dogs inside the house at night (54% agreement), and only 23% believed dogs had a negative impact on wildlife. Cat owners showed the strongest support for confining cats inside the house at night (89% agreement). Cat owners’ non-supportive attitudes towards cat confinement were partly because of higher concern for cat quality of life and lower concern about their cats’ predation behaviours, compared to non-cat owners. The findings provide valuable information to inform more effective strategies to reduce stray animals which would reduce shelter admissions, euthanasia, costs, nuisance issues, potential wildlife predation and negative mental health impacts of euthanasia on staff. Strategies to reduce strays include assisting low-income pet owners to install effective fencing and programs to increase identification. Informing cat owners about bedtime feeding is recommended to assist with night containment, and providing high-intensity free sterilization of owned and semi-owned cats targeted to areas of high cat impoundments is also recommended.

## 1. Introduction

In Australia, dogs are required to be kept securely on the owner’s property and increasingly, state laws or local government by-laws require cats to be confined. However, 60–80% of dogs and cats entering animal welfare shelters, and 80–100% entering municipal facilities (pounds) in Australia are classed as strays [1,2,3,4]. Many dogs but few cats are reclaimed by owners and the proportion varies greatly across local government areas [5]. Some stray dogs and many stray cats are not rehomed and are euthanized. On average, across Australia, approximately 8% of dogs [5] and 30% of cats entering shelters and pounds are euthanized [3,6,7,8], which negatively impacts the mental wellbeing of shelter staff [9,10,11,12,13,14,15]. Modern sheltering practices are increasingly focused on strategies to prevent animals entering shelters and pounds as an effective method to reduce the numbers euthanized [15]. This requires an understanding of the issues associated with stray dogs and cats in urban areas.

Stray dogs in urban areas are owned dogs that have either escaped or were not contained on the owner’s property. For cats, some are escaped owned cats, while others are not contained and have wandered from their property. Cats not contained to the owner’s or carer’s property are often brought by members of the public to the shelter, or are trapped by animal management officers, often in response to complaints. Unlike urban dogs in Australia, some stray cats are semi-owned (fed by people who do not perceive they own the cat) or are unowned cats (obtain food from humans unintentionally) [16,17]. Because most cats entering shelters and pounds are not microchipped or wearing a collar [5], it is difficult to distinguish between owned, semi-owned and unowned domestic cats. However, Australian research suggests that most stray cats are likely owned or semi-owned cats [5,16,17].

Although local government (council) and state laws have for many years required dogs to be confined to an owners’ property, it is only in recent years that many Australian municipal councils have begun introducing regulations aimed at better controlling domestic cats [18,19]. For cats, one of the key drivers for these regulations is a need to address the potential effect of cat predation upon native wildlife populations.

However, many cats are not confined, and studies have reported less support from pet owners for confinement of cats as opposed to dogs. The strongest support is for the containment of dogs on the owner’s property [20,21,22] and inside containment at night for cats [22,23,24,25], but the levels of support have varied over the studies. Low levels of support for total inside containment of cats are reported from both cat owners [22,24,25], non-cat owners [19] and veterinarians [26].

The success of regulations in preventing straying animals relies upon effective implementation, which requires the co-operation of pet owners. Using the theory of planned behavior, co-operation is dependent upon a person’s attitude towards the regulation, how they perceive other peoples’ attitudes to the regulation (subjective norms), and the amount of effort they believe it will take to comply with the regulation (perceived behavioral control) [27]. Additionally, the possibility of being penalized for non-compliance may provide incentive. Societal norms can be of great importance in shaping a person’s attitudes [28].

Despite a widespread belief that pet cats have a negative effect on native wildlife in urban areas [29], there are no well documented effects of pet cats on native wildlife populations [30,31,32,33,34], and many cats do not predate or are not seen to predate, or only predate insects, small lizards or introduced animals such as rats and mice [24,35,36], which may reduce cat owners’ motivation to keep their cat confined.

Lack of compliance with containment legislation may also stem from concerns about the welfare of cats being kept entirely indoors. Certain diseases such as obesity and diabetes are more frequent in indoor only cats, and these reduce lifespan and quality of life of the pet [37,38,39].

The aims of our study were to determine the attitudes and beliefs of Australian dog and cat owners towards confinement of their pets, and how they are influenced by societal norms, gender, and concerns about pet quality of life and wildlife predation. This information might be useful in developing more effective strategies and messaging about confinement to reduce the frequency of stray dogs and cats entering shelters and pounds and being euthanized.

## 2. Materials and Methods

### 2.1. Procedures

A survey was created using online survey software provided by Qualtrics [40]. It requested information on participants’ gender, suburb, age, property type, pet ownership and how many dogs and cats were owned. Participants were asked about whether they observed unowned or feral dogs and cats in their neighborhood. The dependent variables contained questions (4-items) about attitudes toward cat and dog confinement. Participants were asked about the extent to which they believed that pets should be confined inside the house at night, inside the house whenever unsupervised, to an outdoor run when unsupervised and to the owner’s property whenever unsupervised. The survey also included questions on the perception of what other people believe about confinement of dogs and cats (4 items), concerns about the impacts of confinement on pets’ quality of life (QoL) (3 items), concerns about wildlife predation by pets (4 items) and pet–owner bond (2 items). Participants’ attitudes and concerns were assessed by a five-point Likert scale (strongly disagree = 1, disagree = 2, neutral = 3, agree = 4, strongly agree = 5).

Australian pet-related organizations were contacted and asked to assist with distributing the online questionnaire to the general public by publishing a link to it on their webpages, newsletters and/or Facebook pages. These organizations included the RSPCA, Dogs Queensland, Getting to Zero, Cats of Australia and many others. Members of the public used a link to the survey, which allowed them to complete it anonymously if they consented to participate in the survey, and were 18 years or older and an Australian resident. A total of 2183 responses were collected from Australian residents. From these, responses were excluded if the participants did not currently own a dog or a cat (*n* = 80), leaving 2103 eligible participants. We have previously published other components of the questionnaire relating to prey observed to be caught by pet dogs and cats [35].

### 2.2. Data Setup

We divided age in years into three groups: 29 years and below, 30–49 years and 50 years and more, and created three categories for the property types including (a) residence with garden or backyard, (b) residence without garden/backyard, and (c) farm, acreage, semi-rural and rural. Pet ownership status comprised three groups of dog-only owners, cat-only owners and dog and cat owners.

Before computing the items and creating composite measures, we tested the internal consistency between items of each measure (reliability) using Cronbach’s alpha test (α). The Cronbach’s alpha coefficient ranges between 0 and 1, with higher values indicating higher reliability. We also tested the construct validity of each composite measure using Factor Analysis to ensure that the individual items (e.g., on attitudes toward confinement) constructed the concept of interest. Scores for each item of the composite measure were then summed, with higher scores indicating greater levels of agreement to the concept (e.g., confinement, wildlife predation by pets or impact of confinement on pets’ QoL). All composite measures were rescaled to a range from 0 to 1. The reliability of composite measures of attitudes toward cats’ and dogs’ confinement was acceptable (α = 0.66 and 0.81, respectively). Concerns about the impact of containment on pets’ QoL was constructed by summing three items for dogs (α = 0.61) and for cats (α = 0.88) on a similar 5-point scale. To measure others’ supportive attitudes toward pet confinement, we gathered individuals’ perceptions about the extent to which others believe in the confinement of dogs (4 items; α = 0.62) and cats (4 items; α = 0.88). Cat-owner and dog-owner bond were measured using two items each: I regard my cat/dog(s) as a family member, and I am very attached to my cat(s)/dog(s). Concerns about wildlife predation by cats (α = 0.62) and dogs (α = 0.62) were measured using 4 items each. Figure 1 presents the conceptual flowchart of the survey’s data setup and analysis in this study.

### 2.3. Data Analysis

Mean and standard deviation (±SD) were reported for continuous variables and percentage for categorical variables. A series of two-sample t-tests [from summary data] were used to examine if there was a significant mean difference between dog owners’ opinions about dogs, and cat owners’ opinions about cats. To carry out a clear and targeted analysis, we narrowed dog owners’ opinions to dog-related items, and considered cat owners’ responses only to cat-related statements. One-way analysis of variance (ANOVA) and Pearson’s and Spearman’s correlations were used for preliminary analysis (results not shown in tables). To determine the association between demographic variables and agreement to confinement-related items, we recoded each item’s scores 1–3 to 0 (=disagree, reference category) and 4–5 to 1 (=agree), performed binary logistic regression and reported odds ratios (Ors) with 95% confidence intervals (Cis).

To test (1) whether participants’ pet ownership status was related to their attitudes towards confinement of dogs and cats, and (2) whether these associations were mediated by owners’ concerns about the impact of confinement on QOL or concerns about wildlife predation by pets, we performed two path analyses for dogs and cats and investigated possible direct and indirect causal relationships. Multivariable linear regression analyses were performed, and standardized regression coefficients (βs) were reported. For both path diagrams, the Goodness of Fit measures of models confirmed an acceptable fit to the data (RMSEA index was 0.00, which was less than 0.08 required for a good fit. Comparative Fit Index (CFI) was also 1.00, which was more than the recommended threshold 0.90).

Two univariable and multivariate linear regression models were conducted to estimate the crude and adjusted associations between participants’ beliefs and concerns (on wildlife, QoL, others’ attitudes) and participants’ attitudes toward cat and dog confinement. The associations were expressed as the unstandardized regression coefficients (b) with 95% confidence intervals. For all regression analyses, a *p*-value of <0.05 was considered statistically significant. Statistical analyses were carried out using STATA-13.

## 3. Results

### 3.1. Participants’ Demographics

Of the 2103 pet owners, most were female (91%) and aged between 30 to 49 years of age (45%). The age of participants ranged between 18 and 84 (mean and median ≈ 40.0 ± 14.0). The majority lived in Victoria (34%), New South Wales (23%) or Queensland (19%). Most pet owners lived in a property with a garden or backyard (78%), followed by acreage, farm, or rural property (16%). Only 6% of participants lived in a residence that did not have a garden. Wandering or unowned/feral dogs and cats were observed by 9% and 36% of participants, respectively. Of pet owners (*n* = 2103), 1529 owned at least one dog and 1346 owned at least one cat. More than half of dog owners (820/1529 = 54%) owned ≥ 2 dogs. Of cat owners, 56% (758/1346) owned ≥ 2 cats. About 40% of all pet owners owned at least one dog and one cat (779/2013) and 12% of them (249/2103) owned 2 ≥ dogs and 2 ≥ cats. The demographic characteristics of participants can be seen in Table 1.

### 3.2. Supportive Attitudes towards Pet Confinement

#### 3.2.1. General Attitudes

Nearly all dog owners (98%) agreed that dogs should be confined to their property whenever unsupervised (Table 2). In contrast, only 71% of cat owners agreed that cats should be confined to their property whenever unsupervised. Cat owners expressed stronger agreement than dog owners to the other three items: confining inside the house at night (cats: 89% vs. dogs: 54%), inside the house whenever unsupervised (cats: 54% vs. dogs: 18%), and to outdoor run whenever unsupervised (cats: 55% vs. dogs: 32%). In summary, the greatest support was expressed by dog owners for confining dogs to the owners’ property whenever unsupervised (98% agreed/strongly agreed). For cat owners, the greatest support was for confining cats inside the house at night (89% agreement).

#### 3.2.2. Gender of Participants

Gender was a significant factor in determining beliefs about pet containment, when controlled for other demographic factors (e.g., age, property type and pet ownership). Male dog owners were less likely than female dog owners to agree to confining their dogs inside the house at night (OR = 0.5, 95% CI = 0.3,.7) (Table 3). For cat owners, there was no significant association between gender and agreement to the cat confinement items (Table 4).

#### 3.2.3. Age

Age was a robust, independent and significant predictor of attitudes towards dog confinement. Senior dog owners (>50 years of age compared to 29 ≤ years of age) were more supportive of confining dogs inside the house whenever unsupervised (OR = 2.6, 95% CI = 1.7, 3.9), to an outdoor dog run whenever unsupervised (OR = 1.8, 95% CI = 1.3, 2.6) and inside the house at night (OR = 1.6, 95% CI = 1.2, 2.1) (Table 3). For attitudes towards cat confinement, there was no significant difference between age groups, except for the statement “cats should be confined inside the house at night”, to which cat owners above 50 years of age (OR = 1.8, 95% CI = 1.0, 2.9) and those between 30–49 years of age (OR = 1.7, 95% CI = 1.1, 2.7) were more likely to agree than younger cat owners (≤29) (Table 4).

#### 3.2.4. Property Type

After controlling for other demographic factors (e.g., gender, age, pet ownership), type of property remained a significant predictor of attitudes towards dog confinement. Those living on farms, acreages and rural properties had stronger supportive attitudes towards dog confinement than those living in properties with a garden or backyard (*p* < 0.001). They were more likely to agree to confining dogs to the owners’ property (OR = 3.2, 95% CI = 2.3, 4.3) and inside the house whenever unsupervised (OR= 1.4, 95% CI = 2.3, 4.3) (Table 3). For cat owners, those living in properties without a garden or backyard (compared to those with a garden and backyard) were more likely to agree that cats should be confined inside the house at night (OR = 5.1, 95% CI = 1.2, 21.3), inside the house whenever unsupervised (OR = 2.5, 95% CI = 1.5, 4.2) and to the owners’ property whenever unsupervised (OR = 1.9, 95% CI = 1.1, 3.5) (Table 4). In summary, pet owners living in properties with a backyard/garden were less supportive of pet confinement than the other groups.

#### 3.2.5. Number of Dogs/Cats Owned

Owners of two dogs (compared to one dog) were less supportive of confining dogs to an outdoor dog run whenever unsupervised (OR = 0.6, 95% CI = 0.4, 0.8) (Table 3). Owners of two cats, however, were more supportive of this statement about cats (OR = 1.3, 95% CI = 1.0, 1.7). Owners of three or more cats were also more likely to agree to confining cats inside the house whenever unsupervised (OR = 1.5; 95% CI = 1.0, 1.9) and to their property whenever unsupervised (OR = 1.5, 95% CI = 1.0, 2.1) (Table 4). In summary, owning multiple dogs had a negative association with support for dog confinement, while owning multiple cats had a positive effect on supportive attitudes towards cat confinement.

#### 3.2.6. Unowned/Feral Dogs/Cats in Neighborhood

There were positive associations between observing unowned dogs and cats in the neighborhood and support for pet containment. For example, cat owners who observed unowned or feral cats in their neighborhood were also more likely to agree that cats should be confined to their owner’s property whenever unsupervised (OR = 1.4; 95% CI = 1.3, 1.8), or to an outdoor run (OR = 1.5; 95% CI = 1.3, 1.6) (Table 3 and Table 4). Similarly, observing unowned dogs was associated with dog owners’ support for confining dogs to an outdoor run whenever unsupervised (OR = 1.8; 95% CI = 1.2, 2.6). The associations between demographic variables and composite measures of attitudes towards dog and cat confinement were analyzed and visualized (Figure 2). Age, property type and observing unowned dogs were predictors of attitudes towards dog confinement. Property type, having multiple cats and observing unowned cats in the neighborhood were significant predictors of attitudes towards cat confinement.

### 3.3. Pet Owners’ Concerns and Beliefs

#### 3.3.1. Pet–Owner Bond

Nearly all cat and dog owners (98% vs. 97%) strongly agreed or agreed that they were very attached to their pet and regarded their pet as a family member. The pet–owner bond was positively associated with supportive attitudes towards cat confinement (r = 0.17, *p* < 0.001) and dog confinement (r = 0.14, *p* < 0.05), which remained significant after controlling for demographic variables (e.g., gender, age, property type and pet ownership), and pet owners’ opinions and concerns about pet welfare and wildlife (Table 5).

#### 3.3.2. Concerns about Impact of Confinement on Pets Quality of Life (QOL)

Dog and cat owners had minimal concerns about the negative impact on QoL of confining dogs and cats to the owners’ property whenever unsupervised, with 2% of dog owners and 7% cat owners concerned about QoL. However, they were more concerned about negative impacts on QoL associated with confining pets indoors whenever unsupervised, with more dog owners concerned (38%) compared with cat owners (23%; *p* < 0.001). There were similar concerns associated with confining pets to an outdoor run, on QoL (38% dog owners vs. 15% cat owners; *p* < 0.001) (Table 2). Cat owners, generally, were less concerned about cats’ QoL than dog owners were about dogs’ QoL. However, there was a strong correlation between cat owners’ concerns for QoL and their non-supportive attitudes towards cat confinement (r = −0.67, *p* < 0.001). For dog owners, this correlation was lower (r = −0.52, *p* < 0.001) (Figure 3). The multivariable analysis showed that pet welfare concerns remained a robust and significant predictor of non-supportive attitudes towards the confinement of both dogs and cats (Table 5).

#### 3.3.3. Perception of Others’ Attitudes towards Pet Confinement

Nearly 90% of dog owners believed “others” would agree that dogs should be confined to the owners’ property whenever unsupervised, but significantly fewer cat owners (50%) believed “others” supported this for cats (Table 2). For the other three types of confinement, participants perceived “others” to be more supportive of cat confinement than of dog confinement. For example, about 70% of cat owners believed that “others” would agree to confining cats inside the house at night compared to only 25% of dog owners believing “others” supported this for dogs. The least consensus about “others’” perceived agreement for pet confinement was with confining dogs inside the house whenever unsupervised (12% dog owners). More cat owners (41%; *p* < 0.001) perceived that “others” supported this for cats. “Others” were perceived less supportive of confining dogs to an outdoor dog run (23%) than confining cats to an outdoor cat run (41%, *p* < 0.001) (Table 2).

In summary, a comparison between others’ attitudes towards dog and cat confinement suggested that participants believed “others” would agree more to cat confinement than dog confinement (*p* < 0.001), except for confinement to the owners’ property. No significant correlation was found between cat owners’ perceptions of others’ attitudes and their own attitudes towards cat confinement (r = 0.04, *p* = 0.17). However, dog owners’ perceptions of others’ attitudes were significantly positively correlated with their own attitudes towards dog confinement (r = 0.18, *p* < 0.001) (Figure 3). These associations remained unchanged after adjusting for potential covariates (Table 5).

#### 3.3.4. Concerns about Wildlife Predation by Pets

More than half the cat owners (56%) agreed or strongly agreed that cats have a negative impact on native wildlife in their area. This was less than the 23% of dog owners who expressed the same concern about the impact of dogs on native wildlife (*p* < 0.001). Cat owners had stronger beliefs about the contribution of cats to declining native species than dog owners did about dogs (50% vs. 19%, *p* < 0.001). Overall, participants were more concerned about negative impacts of cats on wildlife than dogs. Concerns about wildlife predation by pets were positively associated with supportive attitudes towards cat confinement (r = 0.26, *p* < 0.001) and dog confinement (r = 0.17, *p* < 0.001) (Figure 3), which remained significant after controlling for demographic variables (gender, age, pet ownership, property type), and pet–owner bond and their concerns about pet QoL (Table 5). Bivariate associations between pet owners’ concerns and beliefs and individual items of pet confinement, separately for dogs and cats, are also visualized in Appendix A
Figure A1, Figure A2, Figure A3 and Figure A4.

### 3.4. Pet Ownership Status and Attitudes towards Cat Confinement

A preliminary analysis showed that dog-only owners (compared to cat-only owners) were more inclined to agree with all statements supporting cat confinement, including: cats should be confined to the owner’s property whenever unsupervised (OR = 3.5; 95% CI = 2.5, 4.9), to an outdoor cat run (OR = 3.5; 95% CI = 2.7, 4.6), inside the house whenever unsupervised (OR = 2.8; 95% CI = 2.1, 3.6) and inside the house at night (OR = 2.3; 95% CI = 1.4, 3.7).

We also found that dog-only owners were more concerned about wildlife predation by cats (mean dog-only owners = 0.69 vs. cat-only owners = 0.54; *p* < 0.0001) and less concerned about the impact of confinement on cats’ QoL than cat owners (mean of concerns for dog-only owners = 0.29 vs. cat-only owners = 0.35; *p* < 0.01) (Appendix A
Table A1).

We tested these associations by a path analysis, including a direct effect: cat ownership status (cat-only vs. dog-only) on supportive attitudes towards cat confinement, and two indirect effects: cat ownership status on concerns about cat’s QoL, and concerns about wildlife predation by cats on supportive attitudes toward cat confinement (Figure 4).

For cats, there was a direct and negative relationship between owning a cat and supportive attitudes towards cat confinement (β = −0.21, *p* < 0.001) (Figure 4B). Cat ownership had two [quite equal] indirect effects on owners’ attitudes via concerns about wildlife predation (−0.33 × 0.16 = −0.05) and concerns about QoL (0.11 × −0.58 = −0.06). The total negative effect of owning a cat on attitudes towards cat confinement was β = −0.32, including a direct effect (β = −0.21) and indirect effect (β = −0.11). In summary, our findings suggested that cat owners’ non-supportive attitudes were partly because of their higher concerns about cats’ QoL and lower concern about their cats’ predation behaviors [than non-cat owners].

We also tested the model for dog owners (Figure 4A). Dog ownership had only a direct and negative association with attitudes towards dog confinement (β = −0.16, *p* < 0.001). Non-supportive attitudes of dog owners towards dog confinement were not influenced by their concerns over the dog’s QoL or wildlife predation.

## 4. Discussion

In Australia, most cats and dogs entering animal welfare shelters and municipal facilities are classed as strays [1,2,3,4] and originate from low socioeconomic areas [41,42]. Unfortunately, many of these stray animals are euthanized despite being healthy or treatable, and this negatively impacts the mental health of staff involved, and increases their risk of depression, traumatic stress, substance abuse and suicide [9,10,11,12,13,14]. Modern sheltering practices are increasingly focused on strategies to reduce shelter admissions. This requires an understanding of the issues associated with stray cats and dogs, including pet confinement. The aims of our study were to determine the attitudes and beliefs of Australian cat and dog owners towards four levels of pet confinement, and how they are influenced by social norms, demographic factors, and concerns about pet quality of life and wildlife predation.

### 4.1. Dogs

In our study, dog owners showed the strongest support for confining dogs to the owners’ property, whenever unsupervised, out of the four levels of confinement (98% agreement) (Table 2). This finding is consistent with previous Australian research [20,21,22], and dog owners self-reported a 98–99% compliance rate for confining dogs to their property when unsupervised [20,21]).

Our analysis showed that dog owners’ perceptions of others’ attitudes were significantly positively correlated with their own attitudes towards dog confinement. Nearly 90% of dog owners believed “others” would agree that dogs should be confined to the owners’ property whenever unsupervised. This finding is consistent with previous research in which 95% of dog owners agreed that confinement is a practice that friends and family would agree with [21]. This finding is also consistent with behavioral science research which considers that behaviors are predicted by social norms associated with the behavior (confining dogs to the owners’ property whenever unsupervised is clearly perceived as a social norm by dog owners), in addition to attitudes towards the behavior and perceived personal control in performing the behavior [43,44].

Although our study results are consistent with earlier studies demonstrating a very high level of support for confinement of dogs to the owner’s property when unsupervised, approximately 70% of dogs entering shelters and nearly all dogs impounded by municipal authorities have strayed off the owner’s property [2,5].

Previous research shows that the highest dog intakes into shelters and municipal pounds are from low socioeconomic areas [2,42,45,46,47]. Low socioeconomic status of owners increases the likelihood their dog will stray, be surrendered or be unclaimed because it reflects less household resources [48]. Resource limitations may also inhibit a household’s ability to provide for certain dog-related needs. For example, dog owners with fewer socioeconomic resources may be unable to afford secure dog fencing, or they may be tenants in rental properties who generally have less discretion about how dogs are housed compared to homeowners, or their rental properties may have poor fencing. Dog owners who are tenants also have limited housing choices due to the scarcity of rental properties that allow pets, which can be perceived as a property risk [49].

A reduction in stray dog admissions into shelters and municipal facilities reflects a reduction in the number of stray dogs in the surrounding area. Strategies to assist dog owners to prevent their dog from straying are particularly important in locations of high stray dog intake which are typically low socioeconomic areas. Given that 98% of dog owners supported containing dogs on their property, education about the importance of containment is not likely an effective use of resources, and other methods are indicated. Strategies to consider for reducing stray dog admissions and euthanasia are to assist owners with secure dog fencing and to increase pet dog identification and sterilization rates with free microchipping and sterilization programs targeted to areas of high stray dog impoundments and complaints. In addition, straying dogs should be released to owners as soon as possible, and payment plans should be negotiated for impoundment fees and registration fines, rather than continuing to hold the animal after the owner has been located. Increasing the proportion of pet dogs that are identified with identity tags, collars and microchips facilitates local residents and neighbors as well as veterinarians returning stray dogs directly back to their neighbors, and authorities returning stray dogs back to their owners without impounding [50].

Cost is a barrier to constructing and maintaining containment systems such as dog fencing, and a lack of secure dog fencing contributes to dogs straying, particularly in low socioeconomic areas. Assisting dog owners with building or repairing dog fencing, particularly for dog owners with recurrent stray dog impoundments, and for households in areas of high stray dog intake, proactively reduces stray dog admissions and euthanasia [51].

Dog owners in our study showed less support for confining dogs inside at night with only 54% agreeing or strongly agreeing with this form of containment. Although concern is most often voiced about predation of native wildlife by cats in urban areas, dogs are likely an equal or greater risk to wildlife than cats [35,52,53]. Confinement at night is most effective in protecting native wildlife of conservation concern, because in Australia, most threatened species at risk of potential predation by pet dogs are nocturnal. For example, the endangered koala, vulnerable grey-headed flying fox and threatened bush-stone curlew are all nocturnal [53]. Dog-attack is a key reason for koala admissions to veterinary hospitals in Australia and most attacks occur on the owner’s property [54,55,56,57,58,59]. In 2020–2021, of the 88 threatened native animals rescued due to dog attack in NSW, the vast majority, 86/88 (98%), were nocturnal animals, emphasizing the need to encourage containment of pet dogs indoors or in an enclosure at night in locations where native animals of conservation concern reside [53].

Previous studies have found that positive attitudes toward dog confinement were mostly based on keeping dogs safe, preventing dogs from harming other people and animals, and preventing nuisance [21,22]. The containment of dogs to an owner’s property when unsupervised is required by law in all jurisdictions in Australia, whereas it is required for cats in a minority of jurisdictions (Table A2). Legislation and its enforcement could account for some of the differences between community attitudes towards containment of pet dogs compared to cats. Well-recognized human safety and fatality risks associated with roaming dogs are not applicable to cats, resulting in a greater focus on enforcement of dog containment. In addition, it is far more difficult to keep a cat on an owner’s property with a physical fence compared to a dog because of cats’ climbing ability and agility. Some Australian local governments have concluded that enforcement of cat containment is not feasible because it is time-consuming and cost prohibitive to trap straying cats to identify the owner [60,61,62]. Compounding this, few trapped cats have any identification for owners to be issued infringement notices. The complexities associated with the relationship between humans and the semi-owned stray cat population are another major factor likely influencing attitudes around cat containment. While cat semi-owners feed their stray cats and may provide other care, they are less likely to desex, microchip or contain their cats due to financial limitations (most semi-owners are in low socioeconomic areas) and not perceiving that they own the cat [16,63,64,65]. Other factors are also likely influencing attitudes to the containment of dogs and cats in Australia. For example, dogs but not cats are valued for protection of property. This may also account for the owner’s low support for containment of dogs indoors or in an enclosure at night.

Non-supportive attitudes of dog owners towards dog confinement were not mediated by their concerns over dog quality of life (QoL) or wildlife predation. Only 23% of dog owner participants believed that dogs had a negative impact on wildlife and even fewer (19%) dog owners believed that dogs contributed to a declining native species, despite recent research suggesting that of pets that predate, pet dogs predate a greater proportion of native animals than pet cats [35,52].

Gender was a significant factor in determining beliefs about pet dog containment. Male dog owners were less likely than female dog owners to agree to confining their dogs inside the house at night. Age was also a significant predictor of attitudes towards dog confinement, with senior dog owners (>50 years of age compared to ≤29 years of age) being more supportive of confining dogs inside the house at night. Based on our findings that only 54% of dog owners supported night-time confinement, it is recommended that in specific urban locations where there are threatened nocturnal wildlife that are susceptible to dog predation, education campaigns are targeted to dog owners to increase awareness and motivate owners to contain their dogs inside the house or a run at night to prevent wildlife predation [54,55,56,66].

Given the demographic results, education campaigns should include messaging specifically aimed at male dog owners and young dog owners (under 30 years of age). Communication materials should be distributed containing engaging pictures of species of conservation concern to raise much needed awareness and empathy (a key driver in conservation engagement [67]) among dog owners of the predation risks to wildlife in backyards [68,69]. Increasing awareness and compassion towards wildlife may assist in increasing containment of pet dogs indoors or in enclosures at night to reduce predation of wildlife of conservation concern [54]. Further research is urgently needed to determine the specific urban locations where vulnerable, threatened and endangered species of conservation concern are present that are susceptible to dog predation, so they can be effectively protected with a targeted strategic approach.

### 4.2. Cats

In our study, cat owners showed the strongest support for confining cats inside the house at night of the four levels of confinement (89% agreement [Table 2]). This finding is consistent with previous Australian surveys in which cat owners showed high levels of agreement with confining cats at night [22,23,24,25]. As others have noted, cat owners may have high agreement for night containment because it allows cat owners to balance their beliefs about the positive benefits of containment (preventing injuries, fighting, disease transmission and potential wildlife predation) with their belief that cats need to have periods of non-confinement outside to meet their physical and mental well-being, which cat owners perceive cannot be met in confinement [25,70,71,72,73,74]. Recent research found that containing pet cats at night is the most suitable and achievable behavioural aim for cat owners in relation to reducing potential wildlife predation, because 24-h cat confinement was unlikely to be adopted by many cat owners or supported by veterinarians, whose expert and normative support is critical to change [26]. Indeed, nearly 30% of cat owners in our study did not support it. Social science research [23] and research undertaken by RSPCA NSW found that decisions related to cat containment are most influenced by family members and veterinarians [75].

Cat owners showed stronger support for confining cats inside the house at night (89% agreement) compared to confining cats to their owner’s property whenever unsupervised (71% agreement), which is consistent with previous research [19,22,24,25,74]. However, our study found a higher level of support for confining cats to their owner’s property whenever unsupervised compared to previous studies which reported comparatively lower levels of owner support for 24-h confinement, including 45% agreement [24], 30% agreement [25] and 16% agreement [22]. Our higher level of support (71%) may reflect participants’ interpretation of ‘confining cats to their owner’s property whenever unsupervised’ as not necessarily meaning 24-h containment, because cats could be unconfined ‘when supervised’. It is also possible that the higher level of support reflects changing attitudes over the past five years regarding social norms and pet cat containment, or demographic differences in participants in the different studies. Previous research appears to show a trend towards increasing support over time; however, further research is warranted in this area.

Our study found that cat owners’ non-supportive attitudes towards cat confinement are partly due to lower concern about their cats’ predation behaviors compared to non-cat owners. This finding is consistent with previous research which found that concern about impacts on wildlife was not a significant predictor of containment behavior among cat owners [18,24,25,31,76,77]. This is not surprising, given the low level of predation of native wildlife observed by the same cat owners as in the current study, where a median of three native animals were predated per cat over 6 months, most of which were small lizards (skinks or geckos) [35]. Similarly low levels of predation of native animals by pet cats were reported in suburbs adjacent to bushland in Canberra (median of 1.2 birds/year) [33,78]. However, it is estimated that cat owners only observe 30% of birds, 20% of mammals (of which 98% are introduced mice, rats, rabbits), and 22% of reptiles predated by their pet cat [34,79,80].

Our study also found that cat owners’ non-supportive attitudes towards cat confinement are partly because of higher concern for pet cat quality of life (QoL) compared to non-cat owners. Pet welfare concerns were a robust and significant predictor of non-supportive attitudes towards the confinement of cats (Table 5). This finding is consistent with previous research which found that cat owners’ concerns about the negative impacts on a cat’s physical or mental well-being is an important predictor of confinement [76,81,82,83,84], including indoors-only confinement [82]. Other studies have reported that some cat owners believe that 24-h confinement is cruel or unnatural for cats [70,71,85]. The welfare of cats confined “indoors-only” is an area requiring further research [86].

Unlike dog owners, no significant correlation was found between cat owners’ perceptions of others’ attitudes and their own attitudes towards cat confinement. Our analysis found that only about 50% of cat owners believed “others” would agree that cats should be confined to the owners’ property whenever unsupervised, which was significantly less than dog owners (about 90% believed “others” supported this type of confinement for dogs). About 70% of cat owners believed “others” believed that cats should be confined inside the house at night, compared to only 25% of dog owners believing “others” supported this type of confinement for dogs. Our findings could be explained by the fact that 24/7 confinement of cats to the owners’ property is not currently considered a social norm in Australia, in contrast to dogs. One Australian study found that the lack of support for 24-h cat confinement was consistent across their entire sample, not just in cat owners [19]. However, night cat confinement is increasingly viewed as the norm, and this is reflected in our results.

Although not specifically examined in our analysis, previous research found that cat owners’ perception of their ability to contain their cat (perceived behavioral control) was an important predictor of whether someone fully contained their cat. The perceived ability to contain cats along with concerns about cat mental and physical well-being in confinement were stronger predictors of containment behavior than concern about impacts on wildlife [70,76,81].

Most admissions of free-roaming cats into shelters and municipal facilities are from low socioeconomic areas [45,87,88] where, in Australia, more than 20% of households live on less than $AUD650 per week [89]. Many pet owners in these areas live in rental properties with inadequate fencing or an inability to make property modifications. In the author’s experience (JR), some cat owners unable to contain their cat live in rental properties where summer temperatures exceed 30 °C, but they have no air-conditioning and no screens on windows or doors. While many cat owners successfully contain their cats indoors-only in apartments, which typically would not require expensive infrastructure, for those not living in apartments or who are unable or unwilling to keep their cats indoors-only, cat containment systems can cost in the order of $700–$2000, and for many low-income families or tenants, these costs are simply not feasible.

Night-time containment reduces opportunities for the predation of threatened species most at risk of cat predation in urban areas—nocturnal mammals such as the squirrel glider, eastern pygmy-possum, eastern long-eared bat, little bent-winged bat and grey-headed flying fox [53]. In 2020–2021, of 19 threatened native animals rescued due to cat attack in NSW, the vast majority 16/19 (84%) were nocturnal animals, emphasizing the need for encouraging the night-time containment of pet cats. Bedtime feeding of cats is a highly effective way for cat owners to keep pet cats safely inside at night, especially difficult to contain “door-dasher” cats, and has minimal to no additional cost. Cats are fed inside with half their daily food in the morning, and half just before the owner goes to bed (securing the cat indoors before the evening meal is fed). Only enough food should be fed that can be eaten in 10–15 min, and uneaten food should be removed to prevent obesity and to be most effective for ensuring the cat is contained overnight. Some cats not accustomed to being contained inside become stressed and may be destructive in their escape attempts, or may disrupt the owner’s sleep by scratching at doors or windows. Over time, timid and shy outdoor cats can often adapt to night confinement using bedtime feeding inside. In the author’s experience (JR), providing a litter box is a barrier to night-time containment for some owners and cat-carers. However, cats with healthy gastrointestinal and urinary systems can be contained overnight without a litter box, provided they can toilet outside during the day, as occurs with most dogs.

Our results found that cat owners above 50 years of age and those between 30–49 years of age were more likely to agree than younger cat owners (≤29 years of age) to the statement “cats should be confined inside the house at night”. In addition, cat owners living in properties without a garden or backyard were more likely to agree that cats should be confined inside the house at night (compared to those with a garden or backyard). Education campaigns targeted to cat owners to encourage and promote bedtime feeding should therefore include messaging specifically aimed at young cat owners (under 30 years old) and cat owners with a garden or backyard, given that these demographic groups showed less support for night confinement.

While mandated cat containment, also known as mandated night curfews or 24/7 curfews in Australia and leash laws in the USA, appears to be a logical solution to reduce free-roaming cats, based on evidence, it is not an effective strategy for reducing stray cats [90] and therefore does not reduce associated issues such as nuisance behaviors or potential wildlife predation [8]. Mandated cat containment also has negative consequences including increased cat impoundments persisting for 3 [91] and 20 years [92,93] after introduction. These include increased costs for local governments, increased numbers of healthy but poorly socialized cats killed in shelters and government facilities, and thus increased exposure of staff to the negative mental health impacts, without reducing the overall number of wandering cats or complaint calls [60,91,92,93,94]. Furthermore, mandated cat containment actively prevents resolution of the wandering cat issue because it creates a major barrier to semi-owners (people who feed stray cats they do not perceive they own) taking full ownership of the stray cat they are feeding. Strategies other than mandated containment and trap-adopt or kill are urgently needed to effectively reduce the number of free-roaming urban cats [90]. Most cat admissions into shelters and municipal facilities are of strays born in the preceding 6 to 12 months in low socioeconomic areas [7,87]. Because cost is the main barrier to sterilization, proactive community cat programs based on high-intensity free sterilization of cats targeted to areas of high cat impoundments and shelter admissions (typically low socioeconomic) are recommended to significantly reduce the number of stray cats and impoundments [95,96]. These programs are also very effective at converting semi-owners to owners [97], but mandated containment is a barrier.

It has been proven that attempts at prioritizing wildlife conservation concerns (expressed by non-pet owners) over pet health and welfare concerns (expressed by pet owners) in policymaking produces divisions, conflict and breakdown of meaningful communication between interest groups [83,98,99]. Instead, recognition of the complex needs, values, demographics, and opinions of pet owners, as key partners, is an alternative approach in developing effective animal management strategies [83]. Pet confinement policies are therefore likely to find support among pet owners in Australia if they constructively address owners’ concerns about pet welfare and improve their available resources. Examples of such policies could include providing funding for fencing, education about wildlife predation, teaching owners how to successfully contain cats including using bedtime feeding, and teaching owners how to entertain pets inside. The findings from our research can help to underpin future legislation and guide pet management interventions by targeting various aspects of public opinion about pet confinement to increase the success of the implementation of desired pet management strategies [100].

## 5. Limitations and Future Research

Two main limitations should be considered when interpreting the results of this study. First, data were based on a convenience sampling methodology, meaning results and conclusions drawn from the study may not be accurate if extrapolated to the wider community. Furthermore, the organizations that promoted the survey are likely to have attracted individuals concerned with companion animals and their welfare, which is a potential source of selection bias. Second, despite the large sample size of the study, approximately 91% of participants were female, making it more difficult to generalize findings because the data are more reflective of female attitudes and beliefs. The majority of the participants were female (91%), with males being under-represented (9%) compared to the Australian Bureau of Statistics 2016 census (51% female to 49% male). It is not unusual for more women to respond to attitudinal studies involving animal welfare compared to men [101]. One study found that women respond to online surveys significantly more than men [102], which can be a possible explanation for the overrepresentation of female participants in this study. The gender ratio of participants in this study is important to note, as the bias skewing gender may influence the results. However, women undertake more pet caring behaviors than men [103] and our population of interest was pet owners/carers.

## 6. Conclusions

The results of our study show that the vast majority of dog owners support containing dogs on their property whenever unsupervised (98% agreement), which concurs with previous research. However, approximately 70% of dogs entering shelters and nearly all dogs impounded by municipal authorities have strayed off the owner’s property. Given the high level of support for containment already shown by dog owners, education about the importance of containment is unlikely to be an effective strategy. Stray dogs in urban areas are typically owned dogs that have either escaped or were not being contained on the owner’s property, and most originate from low socio-economic areas indicating that cost barriers and a lack of resources are important contributing factors leading to more stray dogs. Therefore, to reduce the number of stray dogs and related issues including impoundment costs, field services should be assistive rather than punitive, particularly in areas with large numbers of stray dogs.

Dog owners showed less support for confining dogs inside at night (54% agreement) even though confinement at night is most effective in protecting Australian native wildlife of conservation concern at risk of predation by pet dogs. Despite this, only 23% of dog owners believed that dogs had a negative impact on wildlife, and even less (19%) believed that dogs contributed to the decline in native species. Current awareness levels among dog owners of the potential risk dogs pose to native wildlife, especially at night, appears to be low, representing a significant opportunity to raise awareness and reduce wildlife predation. Education campaigns should be targeted to dog owners, including messaging specifically aimed at male dog owners and young dog owners under 30 years of age as these demographic groups showed less support for inside containment at night. The messaging should increase awareness and motivate owners to contain their dogs inside the house or a run at night, particularly in specific urban locations where nocturnal wildlife of conservation concern are present, to reduce their risk of predation [54]. Further research is urgently needed to determine the specific urban locations where species of conservation concern are present, so they can be effectively protected with a targeted strategic approach.

The results of our study found that cat owners show the strongest support for confining cats inside the house at night (89% agreement), which concurs with previous research. Containing pet cats at night is the most suitable and achievable behavioral aim for cat owners to keep cats safe and reduce potential wildlife predation and is increasingly viewed as the norm. Furthermore, night confinement is most effective in protecting Australian native wildlife of conservation concern at risk of predation by pet cats, which are mainly nocturnal mammals. Total confinement inside may not be adopted by cat owners due to concerns about cat welfare, less concern about wildlife predation, property limitations and cost barriers. Bedtime feeding should be promoted to encourage the confinement of cats at night.

In summary, key findings from this study provide valuable information for developing more effective strategies and messaging about pet confinement to reduce the number of stray cats and dogs impounded and euthanized, thereby reducing negative mental health impacts on staff, and related burnout and high attrition rates. More effective strategies would also reduce issues such as the impact of nuisance behavior or potential wildlife predation.

## Figures and Tables

**Figure 1 animals-13-01067-f001:**
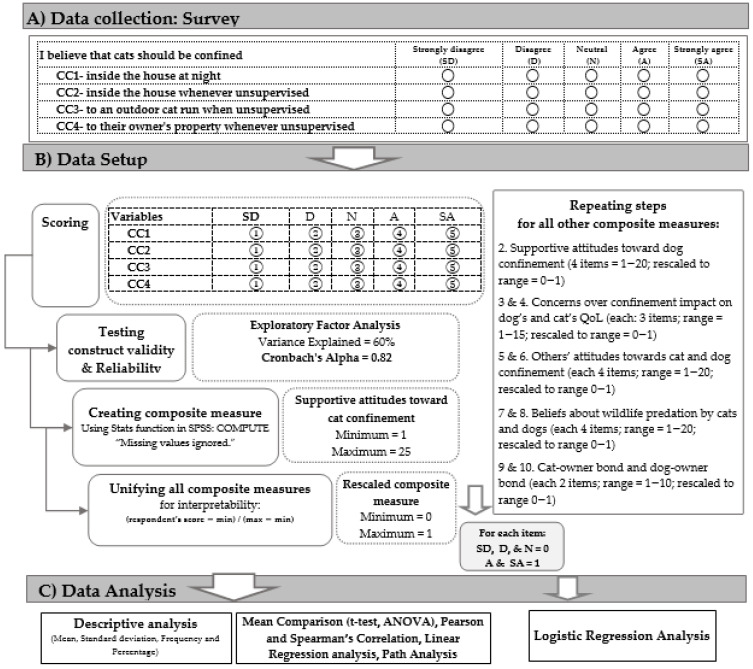
Conceptual flowchart of survey data setup and analysis: (**A**) data collection using an online survey with 5 scale items about cat and dog confinement; (**B**) data cleaning, scoring, combining, validating, and creating composite scales for further statistical analysis; (**C**) data analysis including descriptive, univariable and multivariable analysis methods.

**Figure 2 animals-13-01067-f002:**
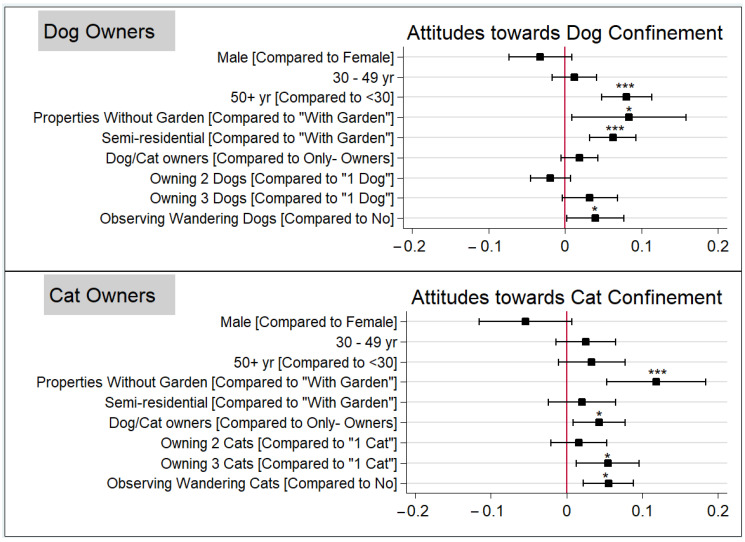
Associations between demographic variables and attitudes towards dog and cat confinement; * indicates significant differences between categories * *p* < 0.05, *** *p* > 0.001. Positive values on the x axis indicate greater support for containment and negative values indicate less support.

**Figure 3 animals-13-01067-f003:**
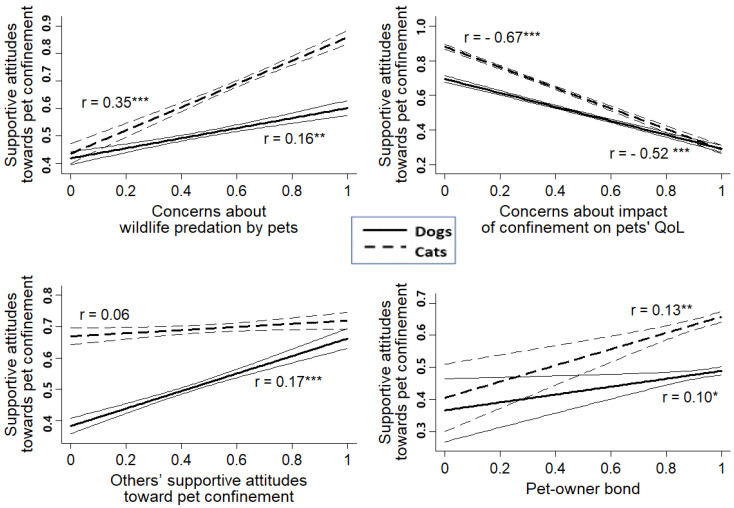
Bivariate associations between supportive attitudes towards pet confinement and pet owners’ concerns about wildlife and pets’ QoL, perception of others’ attitudes and pet–owner bond, separately for dogs and cats; Pearson’s correlation coefficients (r) are presented for each line with significance level denoted by the asterisk (* *p* < 0.05, ** *p* < 0.01, *** *p* > 0.001).

**Figure 4 animals-13-01067-f004:**
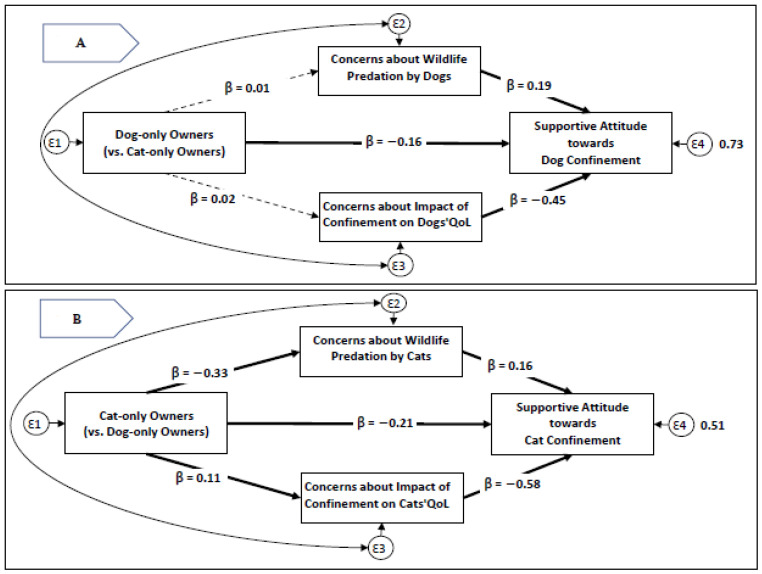
Path analysis for the relationships between pet ownership and attitudes towards pet confinement for dogs (**A**) and cats (**B**), testing mediating effects of Concerns about impacts of confinement on pet’s QoL and Concerns about wildlife predation by pets; values are standardized regression coefficients (Beta); solid arrows represent statistically significant associations (*p* < 0.05). Dashed arrows indicate non-significant associations (*p* > 0.05).

**Table 1 animals-13-01067-t001:** Demographic characteristics of pet owners (*n* = 2103).

	N (%)
State	
VIC	724 (34.4%)
NSW	478 (22.7%)
QLD	391 (18.6%)
ACT	240 (11.4%)
WA	115 (5.5%)
TAS	78 (3.7%)
SA	66 (3.1%)
NT	11 (0.5%)
Gender	
Female	1905 (90.6%)
Male	198 (9.4%)
Age	
29 years and below	577 (27.4%)
30–49 years	939 (44.7%)
50 years and above	587 (27.9%)
Property type	
Residence with garden/ backyard	1638 (78.0%)
Residence without garden/backyard	124 (6.0%)
Acreage, farm, semi-rural, rural	339 (16.1%)
Pet ownership	
Only dog	757 (36.0%)
Only cat	574 (27.3%)
Both dog and cat	772 (36.7%)
How many dogs owned	
none	574 (27.3%)
1	709 (33.7%)
2	584 (27.8%)
3+	236 (11.2%)
How many cats owned	
none	757 (36.0%)
1	588 (28.0%)
2	448 (21.3%)
3+	310 (14.7%)
Wandering (unowned/feral) dogs in neighborhood	
Yes	194 (9.2%)
Wandering (unowned/feral) cat in neighborhood	
Yes	758 (36.0%)
Owner–pet bond (agree and strongly agree)	
Regard dog as a family member	1365 (98.6%)
Attached to pet dog	1377 (99.4%)
Regard cat as a family member	1130 (97.5%)
Attached to pet cat	1142 (98.4%)

**Table 2 animals-13-01067-t002:** Comparison of dog owners’ (*n* = 1529) and cat owners’ (*n* = 1346) responses to statements regarding attitudes and opinions towards confinement of their pet dogs and cats.

Individual Statement	Response Options %					Mean (SD) ^a^ (Range 1–5)	Mean Comparison ^b^	Composite Measure ^c^	Mean (SD) ^a^ (Range 0–1)		Mean Comparison ^b^
	SD (1)	D (2)	N (3)	A (4)	SA (5)						
I believe that…								Supportive attitudes towards pet confinement	Dogs Cats	0.48 (0.22) 0.64 (0.28)	] *p* < 0.001
dogs should be confined to their owner’s property whenever unsupervised	<0.5%	<0.5%	1%	11%	87%	4.86 (0.40)	] *p* < 0.001				
cats should be confined to their owner’s property whenever unsupervised	2%	12%	15%	27%	44%	4.00 (1.11)					
dogs should be confined inside the house at night	4%	20%	21%	25%	29%	3.54 (1.20)	] *p* < 0.001				
cats should be confined inside the house at night	<1%	4%	6%	19%	70%	4.55 (0.82)					
dogs should be confined inside the house whenever unsupervised	11%	48%	23%	10%	8%	2.55 (1.07)	] *p* < 0.001				
cats should be confined inside the house whenever unsupervised	3%	23%	20%	20%	34%	3.50 (1.25)					
dogs should be confined to an outdoor dog run when unsupervised	11%	35%	23%	21%	11%	2.72 (1.16)	] *p* < 0.001				
cats should be confined to an outdoor cat run when unsupervised	4%	18%	23%	27%	28%	3.57 (1.19)					
I believe that other people agree that…								Others’ supportive attitude towards pet confinement	Dogs Cats	0.42 (0.18) 0.50 (0.25)	] *p* < 0.001
dogs should be confined to their owner’s property whenever unsupervised	<1%	2%	9%	35%	53%	4.38 (0.79)	] *p* < 0.001				
cats should be confined to their owner’s property whenever unsupervised	2%	19%	25%	33%	21%	3.52 (1.05)					
dogs should be confined inside the house at night	4%	33%	38%	21%	4%	2.89 (0.94)	] *p* < 0.001				
cats should be confined inside the house at night	1%	13%	20%	38%	28%	3.84 (1.00)					
dogs should be confined inside the house whenever unsupervised	11%	47%	30%	9%	3%	2.45 (0.90)	] *p* < 0.001				
cats should be confined inside the house whenever unsupervised	2%	28%	29%	27%	14%	3.21 (1.05)					
dogs should be confined to an outdoor dog run whenever unsupervised	7%	37%	33%	19%	4%	2.74 (0.96)	] *p* < 0.001				
cats should be confined to an outdoor cat run whenever unsupervised	2%	24%	32%	29%	12%	3.24 (1.01)					
I believe that confining…								Concerns over confinement impact on pet’s QoL	Dogs Cats	0.47 (0.27) 0.33 (0.30)	] *p* < 0.001
dogs to their property when unsupervised reduces their QoL	64%	30%	4%	1%	1%	1.44 (0.70)	] *p* < 0.001				
cats to their property when unsupervised reduces their QoL	44%	38%	11%	5%	2%	1.84 (0.95)					
dogs indoors when unsupervised reduces their QoL	17%	26%	19%	27%	11%	2.91 (1.28)	] *p* < 0.001				
cats indoors when unsupervised reduces their QoL	34%	31%	12%	16%	7%	2.23 (1.29)					
dogs to a dog run when unsupervised reduces their QoL	13%	27%	22%	28%	10%	2.94 (1.21)	] *p* < 0.001				
cats to a cat run when unsupervised reduces their QoL	34%	35%	16%	11%	4%	2.17 (1.14)					
I believe that…								Concerns about wildlife predation by pets	Dogs Cats	0.46 (0.22) 0.57 (0.21)	] *p* < 0.001
dogs have a negative impact on the native wildlife population in my area	22%	32%	22%	15%	8%	2.54 (1.21)	] *p* < 0.001				
cats have a negative impact on the native wildlife populations in my area	8%	16%	20%	36%	20%	3.46 (1.20)					
dogs contribute to declining numbers of some native wildlife species in my area	20%	35%	26%	16%	3%	2.50 (1.07)	] *p* < 0.001				
cats contribute to declining numbers of some native wildlife species in my area	8%	18%	24%	32%	18%	3.34 (1.19)					

(^a^) Mean is the average of scores and calculated by {Sum of scores} / {Total numbers of participants}; (^b^) two sample t-tests [from summary data] were used to examine if there was a significant mean difference between dog owners’ opinions about dogs and cat owners’ opinions about cats; (^c^) composite measures were created by combining relevant statements separately for cats and dogs, rescaled to range from 0–1 for interpretability; Abbreviations = SD: Strongly disagree, D: Disagree, N: Neutral, A: Agree, SA: Strongly agree; QoL: Quality of Life; M: Mean; SD: Standard deviation; Dog owners included those who own only dogs and both dogs and cats; Cat owners included those who own only cats and both dogs and cats; Responses of dog owners to items about cats and vice versa were not taken into consideration.

**Table 3 animals-13-01067-t003:** Associations between demographic variables and agreement (agree/strongly agree) to statements about dog confinement as well as composite measure “supportive attitudes towards dog confinement” among dog owners (*n* = 1529).

	Individual Statements ^a^								Composite Measure ^b^	
	Confined Inside the House at Night		Confined Inside the House Whenever Unsupervised		Confined to an Outdoor Dog Run Whenever Unsupervised		Confined to Owner’s Property Whenever Unsupervised		Supportive Attitudes Towards Dog Confinement	
	*n* (%)	Adjusted OR (95% CI)	*n* (%)	Adjusted OR (95% CI)	*n* (%)	Adjusted OR (95% CI)	*n* (%)	Adjusted OR (95% CI)	Mean (SD)	Adjusted b (95% CI)
Gender										
Female (ref)	673 (55%)	1	222 (18%)	1	324 (27%)	1	1199 (99%)	1	0.49 (0.22)	0
Male	55 (40%)	**0.5 (0.3, 0.7)**	18 (13%)	0.65 (0.4, 1.1)	46 (34%)	1.2 (0.8, 1.9)	135 (99%)	1.5 (0.18, 13.2)	0.47 (0.23)	−0.03 (−0.08, 0.01)
Age										
29 years and below (ref)	173 (50%)	1	42 (12%)	1	79 (23%)	1	341 (99%)	1	0.45 (0.21)	0
30–49 years	315 (52%)	1.0 (0.8, 1,4)	95 (16%)	1.3 (0.9, 2)	140 (23%)	0.9 (0.6, 1.3)	607 (100%)	4.3 (0.8, 22.9)	0.46 (0.22)	0.01 (−0.02, 0.04)
50 years and above	240 (61%)	**1.6 (1.2, 2.1)**	103 (26%)	**2.6 (1.7, 3.9)**	151 (39%)	**1.8 (1.3, 2.6)**	386 (99%)	1.2 (0.3, 4.4)	0.54 (0.23)	**0.08 (0.05, 0.11)**
Property type										
Residence with garden/ backyard (ref)	523 (53%)	1	156 (16%)	1	207 (21%)	1	981 (99%)	1	0.47 (0.22)	0
Residence without garden/backyard	22 (65%)	1.9 (0.9, 3.9)	9 (27%)	**2.3 (1.1, 5.2)**	7 (21%)	.9 (.4,2.2)	34 (100%)	-	0.54 (0.20)	**0.08 (0.01, 0.15)**
Acreage, farm, semi-industrial/rural	154 (57%)	1.1 (0.8, 1.5)	63 (23%)	**1.4 (1.0, 2.1)**	129 (48%)	**3.2 (2.3, 4.3)**	263 (97%)	**0.3 (0.1, 0.9)**	0.55 (0.23)	**0.06 (0.03, 0.09)**
Pet ownership										
Only dog (ref)	317 (53%)	1	99 (17%)	1	151 (25%)	1	593 (99%)	1	0.47 (0.22)	0
Both dog and cat	382 (55%)	1.1 (0.8, 1.3)	129 (19%)	1.1 (0.8, 1.5)	192 (28%)	1.1 (0.8, 1.4)	686 (99%)	1.7 (0.5, 5.8)	0.49 (0.22)	0.02 (−0.01, 0.04)
How many dog[s] owned										
1 (ref)	320 (53%)	1	96 (16%)	1	165 (28%)	1	592 (99%)	1	0.48 (0.21)	0
2	261 (52%)	0.9 (0.7, 1.2)	86 (17%)	1.0 (0.7, 1.4)	111 (22%)	0.6 (0.4, 1.1)	495 (99%)	2.6 (0.6, 10.4)	0.47 (0.22)	−0.02 (−0.05, 0.01)
3+	118 (61%)	1.3 (9, 1.8)	46 (24%)	1.3 (0.9, 2.1)	67 (35%)	0.9 (0.6, 1.4)	192 (99%)	3.9 (0.4, 33.2)	0.54 (0.23)	0.03 (−0.01, 0.1)
Unowned/feral dog in neighborhood										
No (ref)	638 (54%)	1	211 (18%)	1	170 (24%)	1	700 (99%)	1	0.48 (0.22)	0
Yes	89 (55.3%)	1.0 (0.7, 1.4)	29 (18%)	0.8 (0.5, 1.3)	196 (31%)	**1.8 (1.2, 2.6)**	618 (98%)	0.2 (0.1, 0.8)	0.54 (0.22)	**0.04 (0.0, 0.1)**

^a^ To predict agreement to each statement (disagree = 0; agree = 1), a series of multivariable logistic regression analyses were performed and odds ratios (OR) with 95% confidence intervals (CI) in parentheses reported; ^b^ for the composite measure (range 0–1), a multivariable regression analysis was performed and unstandardized estimates; (b) with 95% confidence intervals (CI) in parentheses reported; variables in each model were controlled for each other; odds ratios in bold are statistically significant.

**Table 4 animals-13-01067-t004:** Associations between demographic variables and agreement (agree/strongly agree) to statements about cat confinement as well as composite measure “supportive attitudes towards cat confinement” among cat owners (*n* = 1289).

	Individual Statements ^a^								Composite Measure ^b^	
	Confined Inside the House at Night		Confined Inside the House Whenever Unsupervised		Confined to an Outdoor Cat Run Whenever Unsupervised		Confined to Owner’s Property Whenever Unsupervised		Supportive Attitudes Towards Cat Confinement	
	*n* (%)	Adjusted OR (95% CI)	*n* (%)	Adjusted OR (95% CI)	*n* (%)	Adjusted OR (95% CI)	*n* (%)	Adjusted OR (95% CI)	Mean (SD)	Adjusted b (95% CI)
Gender										
Female (ref)	987 (90%)	1	600 (54%)	1	619 (56%)	1	794 (72%)	1	0.65 (0.27)	0
Male	84 (88%)	0.7 (0.4, 1.4)	49 (52%)	0.9 (0.5, 1.4)	48 (51%)	0.7 (0.5, 1.2)	63 (66%)	0.7 (0.5, 1.2)	0.59 (0.29)	−0.05 (−0.11, 0.01)
Age										
29 years and below (ref)	261 (86%)	1	157 (52%)	1	169 (56%)	1	212 (70%)	1	0.62 (0.28)	0
30–49 years	500 (91%)	**1.7 (1.1, 2.7)**	314 (57%)	1.2 (0.9, 1.6)	311 (56%)	1.0 (0.7, 1.3)	396 (72%)	1.1 (0.8, 1.6)	0.65 (0.27)	0.03 (−0.01, 0.06)
50 years and above	310 (91%)	**1.8 (1.0, 2.9)**	178 (52%)	1.0 (0.7, 1.4)	187 (54%)	0.9 (0.7, 1.3)	249 (73%)	1.1 (0.8, 1.6)	0.65 (0.28)	0.03 (−0.01, 0.01)
Property type										
Residence with garden/ backyard (ref)	792 (90%)	1	461 (52%)	1	468 (53%)	1	610 (69%)	1	0.63 (0.28)	0
Residence without garden/backyard	76 (97%)	**5.1 (1.2, 21.3)**	55 (71%)	**2.5 (1.5, 4.2)**	52 (67%)	**2.0 (1.2, 3.4)**	62 (79%)	1.9 (1.1, 3.5)	.071 (0.25)	**0.12 (0.05, 0.18)**
Acreage, farm, semi-industrial/rural	155 (83%)	**0.5 (0.3, 0.7)**	100 (54%)	1.0 (0.7, 1.3)	112 (60%)	1.2 (0.9, 1.7)	144 (77%)	**1.3 (0.9, 1.9)**	0.68 (0.28)	0.02 (−0.02, 0.06)
Pet ownership										
Only cat (ref)	416 (89%)	1	243 (52%)	1	250 (53%)	1	327 (70%)	1	0.61 (0.29)	0
Both dog and cat	607 (89%)	1.3 (0.9, 1.9)	373 (55%)	1.2 (0.9, 1.5)	382 (56%)	1.1 (0.8, 1.4)	489 (72%)	1.0 (0.8, 1.4)	0.66 (0.27)	**0.04 (0.01, 0.08)**
How many cat[s] owned										
1 (ref)	456 (90%)	1	261 (52%)	1	249 (49%)	1	348 (68%)	1	0.62 (0.27)	0
2	332 (88%)	0.9 (0.6, 1.3)	198 (53%)	1.1 (0.8, 1.4)	207 (55%)	**1.3 (1.0, 1.7)**	267 (71%)	1.2 (0.9, 1.6)	0.63 (0.28)	0.02 (−0.02, 0.05)
3+	235 (90%)	0.9 (0.6, 1.6)	157 (60%)	**1.5 (1.0, 1.9)**	176 (67%)	**2.1 (1.6, 2.9)**	201 (77%)	**1.5 (1.0, 2.1)**	0.69 (0.28)	**0.05 (0.01, 0.10)**
Unowned/feral cat in neighborhood										
No (ref)	640 (89%)	1	739 (53%)	1	378 (53%)	1	495 (69%)	1	0.66 (0.26)	0
Yes	431 (90%)	1.2 (0.8, 1.8)	270 (56%)	1.1 (0.9, 1.5)	289 (60%)	**1.3 (1.0, 1.6)**	362 (76%)	**1.4 (1.0, 1.8)**	0.74 (0.26)	**0.06 (0.02, 0.09)**

^a^ To predict agreement to each statement (disagree = 0; agree = 1), a series of multivariable logistic regression analyses were performed and odds ratios (OR) with 95% confidence intervals (CI) in parentheses reported; ^b^ For the composite measure (range 0–1), a multivariable regression analysis was performed and unstandardized estimates (b) with 95% confidence intervals (CI) in parentheses reported; variables in each model were controlled for each other; odds ratios in bold are statistically significant.

**Table 5 animals-13-01067-t005:** Prediction of supportive attitudes towards cat and dog confinement by pet owners’ concerns about wildlife and pets’ QoL, perception of others’ attitudes and pet–owner bond, separately for dogs and cats.

	Supportive Attitudes Towards Cat Confinement Among Cat Owners (*n* = 1346)	Supportive Attitudes Towards Dog Confinement Among Dog Owners (*n* = 1529)
Concerns over confinement impact on pet’s QoL	**−0.59 (−0.62, −0.54)**	**−0.40 (−0.44, −0.36)**
Concerns about wildlife predation by pet	**0.21 (0.15, 0.26)**	**0.13 (0.08, 0.18)**
Perception of others’ attitudes towards pet confinement	0.05 (−0.01, 0.10)	**0.18 (0.12, 0.23)**
Pet–owner bond	**0.17 (0.08, 0.25)**	**0.14 (0.05, 0.22)**
**Adjusted R^2^**	**0.48**	**0.33**

Note: Cell entries are unstandardized estimates (b) from two multivariable linear regression analyses (separately for dogs and cats), with 95% confidence intervals in parentheses; coefficients are controlled for gender, property type, number of pets owned, age, pet ownership status and feral/stray cats/dogs in the neighborhood; results for controlled variables are not presented; coefficients in bold are statistically significant at 0.05 level; **Abbreviations**: QoL: Quality of Life.

## Data Availability

The raw data supporting the conclusions of this article will be made available by the authors, without undue reservation.

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
