# Peer review of "Attitudes and Beliefs of a Sample of Australian Dog and Cat Owners towards Pet Confinement"

_animals, 2023, doi:10.3390/ani13061067_

Round 1

Reviewer 1 Report

The manuscript presents the findings of a dog and cat owner questionnaire exploring attitudes to pet containment. Responses were collected from a convenience sample of 1,539 dog owners and 1,346 cat owners from across Australia. Attitudes to four different types of containment and how these were influenced by social norms, demographic factors and concerns about wildlife and pet quality of life were investigated. Important differences were found between dog and cat owners. Dog owners were widely supportive of containment to owner property. Cat owners were also largely supportive of containment both overnight and to owner property when unsupervised, however this was influenced by cat owners’ concern for cat quality of life and level of concern about their cat’s predatory behaviour. Overall, this is an interesting and well-presented study on an important topic. The Discussion and conclusions drawn from the results show considerable bias, as does the selection of literature cited. In addition, much of the cited literature is not peer-reviewed. These issues need to be rectified.

Introduction

74 – remove ‘perceived’.

90 – Instead of referring to a ‘widespread belief’ please cite relevant literature e.g.

Legge, S., Woinarski, J.C., Dickman, C.R., Murphy, B.P., Woolley, L.A. and Calver, M.C., 2020. We need to worry about Bella and Charlie: the impacts of pet cats on Australian wildlife. Wildlife Research47(8), pp.523-539.

The literature that is cited (reference 29), is a self-citation and does not reflect the proportion of a pet cat’s catch that is not witnessed by their owner. In addition, as mentioned in the Discussion (line 528), this same study reported pet cats predating native reptiles.

91 – Please replace ‘many’ with ‘some’. Previous studies have shown that most pet cats will predate if given the opportunity e.g.

Bruce, S.J., Zito, S., Gates, M.C., Aguilar, G., Walker, J.K., Goldwater, N. and Dale, A., 2019. Predation and risk behaviors of free-roaming owned cats in Auckland, New Zealand via the use of animal-borne cameras. Frontiers in Veterinary Science6, p.205.

MATERIALS AND METHODS

107 – replace ‘respondents’ with ‘participants’. Or use one or the other consistently throughout.

RESULTS

215 – replace ‘regardless’ with ‘when controlled for’

279 – remove the ‘1.0’ where this appears twice in the confidence intervals

300 – replace ‘minimum’ with ‘minimal’

DISCUSSION

427 – rephrase sentence e.g. ‘Low socioeconomic status of owners increases the likelihood their dog will stray, be surrendered or be unclaimed’. Also, correct formatting of this citation.

438 – replace ‘reduce stray dogs’ with ‘prevent their dog from straying’

453 – replace ‘stray dogs’ with ‘dogs straying’

461-464 – needs citations from peer-reviewed literature. These claims need to be appropriately referenced or removed.

467 – please include discussion of the role of legislation on influencing social norms around containment for dogs versus cats. Containment of dogs to owner’s property is required by law in all jurisdictions in Australia, whereas it is required for cats in a small minority of jurisdictions. This could account for some of the differences in your findings between dogs and cats and also the low levels of agreement with containment of dogs indoors or in enclosures.

479 – please provide a justification for recommending dogs be contained indoors or in enclosures. Is there any evidence from the literature that dogs predate wildlife on their owner’s property? Or is this a hypothetical risk?

486 – ‘Communication materials should be distributed containing engaging pictures of species of conservation concern.’ Please remove or provide a justification for this statement.

504 – Please cite peer-reviewed literature e.g.

Elliott, A., Howell, T.J., McLeod, E.M. and Bennett, P.C., 2019. Perceptions of responsible cat ownership behaviors among a convenience sample of Australians. Animals9(9), p.703.

526 – please mention the proportion of a cat’s prey that is brought home, which has been estimated at around 15% of their total kill. In which case, the study cited (reference 29) actually reported a substantially higher toll on native wildlife per cat than has been reported in national estimates e.g.

Legge, S., Woinarski, J.C., Dickman, C.R., Murphy, B.P., Woolley, L.A. and Calver, M.C., 2020. We need to worry about Bella and Charlie: the impacts of pet cats on Australian wildlife. Wildlife Research47(8), pp.523-539.

557 – please use more objective and balanced language in this paragraph with appropriate citations. Cats can also be contained indoors only, this is practiced by many people living in apartments and does not require expensive infrastructure.

560 – this paragraph is not relevant. Please remove.

582 – this paragraph demonstrates a strong anti-containment bias and should be removed or re-written with more objective and balanced language. Citations should be of peer-reviewed literature.

CONCLUSIONS

Conclusions need to be made more concise and need to reflect the findings of the study, especially with regards to cat owners.

645 – this paragraph is not necessary

662 – This statement should be removed. Your results also showed high levels of agreement by cat owners with containment to owner property, not just with containment overnight.

664 – this statement should also be removed. It is not supported by the majority of the contemporary literature.

Reviewer 2 Report

This is an excellent and well-constructed paper.  I have no suggestions for improvement.

Author Response

We are very grateful to you for taking the time to read our manuscript and for your supportive comments.

Reviewer 3 Report

Page 3 the most significant problem with this survey is the skewed nature of the data as explained by the authors later - as the data is self selecting, it has bias towards pet owners, females and certain States eg S Australia is poorly represented per capita compared to ACT.  More needs to assess this looking at other surveys with blind sampling and assessing the differences. 

Page 16 there needs to be more discussion around the legislation and it's impact eg all States make it an offence to abandon or have a dog straying and some an offence for the cat to be out.  There needs to be a table outlining what the legislation is.  The impact of this on people's views also is required eg was this a question in the survey. 

Previous studies showed the link between income and straying behaviour but this is not answered in the discussion - was there a question on income and could this be extrapolated on behaviours.

The study adds to the literature on this issue but is severely limited by the self sampling and the biases that result - - it needs to be rewritten to take this into account in the discussions as this limits it's scientific validity. 

This is an addition to the previous studies  

Round 2

Reviewer 1 Report

Thank you for your detailed and considered responses and well done on an interesting study.